# Mechanism of Potassium Release from Feldspar by Mechanical Activation in Presence of Additives at Ordinary Temperatures

**DOI:** 10.3390/ma17010144

**Published:** 2023-12-27

**Authors:** Xianmei Zhang, Zhenquan He, Wei Jia, Fanrong Meng, Wenju Zhang, Changai Lu, Xiangyang Hao, Guosheng Gai, Zhaohui Huang, Minggang Xu, Kaijun Wang, Sining Yun

**Affiliations:** 1State Key Laboratory of Efficient Utilization of Arid and Semi-Arid Arable Land in Northern China, Institute of Agricultural Resources and Regional Planning, Chinese Academy of Agricultural Sciences, Beijing 100081, China; zhangwenju01@caas.cn (W.Z.); luchangai@caas.cn (C.L.); xuminggang@caas.cn (M.X.); 2Key Laboratory of Arable Land Quality Monitoring and Evaluation, Ministry of Agriculture and Rural Affairs, Institute of Agricultural Resources and Regional Planning, Chinese Academy of Agricultural Sciences, Beijing 100081, China; 3Wuxi Research Institute of Applied Technologies, Tsinghua University, Wuxi 214100, China; mengfr00@sina.com (F.M.); gaigs@139.com (G.G.); 4Cultivated Land Quality Monitoring and Protection Center, Ministry of Agriculture and Rural Affairs of China, 20 Maizidian Street, Beijing 100125, China; jiawei711@126.com; 5Engineering Research Center of Ministry of Education for Geological Carbon Storage and Low Carbon Utilization of Resources, Beijing Key Laboratory of Materials Utilization of Nonmetallic Minerals and Solid Wastes, National Laboratory of Mineral Materials, School of Material Sciences and Technology, China University of Geosciences, Beijing 100083, China; haoxy@cugb.edu.cn (X.H.); huang118@cugb.edu.cn (Z.H.); 6Shanxi Province Key Laboratory of Soil Environment and Nutrient Resources, Engineer and Technology Academy of Ecology and Environment, Shanxi Agricultural University, Taiyuan 030031, China; 7Functional Materials Laboratory (FML), School of Materials Science and Engineering, Xi’an University of Architecture and Technology, Xi’an 710055, China; wangkaijun@xauat.edu.cn (K.W.); yunsining@xauat.edu.cn (S.Y.)

**Keywords:** potassium feldspar powder, mechanochemical activation, water soluble potassium, available potassium, potassium availability

## Abstract

To improve the potassium availability of feldspar at ordinary temperatures, the mechanical grinding and addition of sodium hydroxide/salts were employed to study the effects of mechanical activation and strong alkali addition on particle characteristics, water-soluble potassium, and the available potassium of feldspar. A laser particle size analyzer was utilized for the direct determination of particle size distribution (PSD) using ground samples. The Brunauer–Emmett–Teller (BET) method was employed for specific surface areas. X-ray diffraction (XRD) was employed for structural characterization, scanning electron microscopy (SEM) for morphology exploration, and energy dispersive spectroscopy (EDS) to determine the chemical composition of potassium feldspar powder. The results revealed that the mechanical activation of potassium feldspar could reduce the particle size and produce agglomerated nanoparticles in the later period. The addition of NaOH and sodium salt did not cause agglomeration, and NaOH dissolved the nanoparticles. The water-soluble potassium content of feldspar in each treatment increased during mechanical grinding, from 21.64 mg kg^−1^ to 1495.81 mg·kg^−1^, by adding NaOH 5% weight of potassium feldspar powder and to 3044.08 mg·kg^−1^ by adding NaOH 10% weight with effects different from those of mechanical shaking. By comparison, only 162.93 mg·kg^−1^ water-soluble potassium was obtained by adding NaOH 5% weight. The dissolved potassium in the former case was significantly higher than in the latter, and the addition of NaOH and sodium salts significantly enhanced the water-soluble potassium contents due to ion exchange. Furthermore, the addition of sodium hydroxide improved the water-soluble potassium due to its mechanochemical action on potassium feldspar. The mechanical energy changed the crystal structure of potassium feldspar, explaining the increase in available potassium. The addition of sodium salts did not promote change in the feldspar’s structure, thereby did not raise the available potassium content. The reason for this was related to the mechanochemical action on sodium hydroxide and feldspar, which could promote the dissolution of fine particles, thereby incrementing the available potassium.

## 1. Introduction

Potassium (K) is an important element for biological functions [1], and its demand is growing rapidly owing to its good prospects in China, India, Brazil, and Indonesia, as well as a rebound in Belarus [2]. The potassium salts are mainly used to produce potash fertilizers. Currently, potash resources are abundant worldwide, but resource distribution and production layout are extremely unbalanced [3]. Since the production is controlled by a few countries, shortage of potash fertilizer still exists in most consumption areas [3]. Feldspar occupies about 50% of the Earth’s crust mass and 64% of the Earth’s crust volume [4,5]. Potassium feldspar has been widely studied as a potential potash fertilizer resource [6,7,8]. It may be alternated by water in some conditions [9] and used to produce multiphase slow-release fertilizers by sintering reaction [10]. 

However, most studies encountered difficulties due to the stable framework structure and chemical properties of potassium feldspar [11], thereby it cannot be directly absorbed by plants. Since the structure is connected by silicon (aluminum) and oxygen tetrahedrons through oxygen bridging in three dimensions, it cannot be easily decomposed by acid or alkali at normal temperatures and pressures. As a result, many processes have been used to extract potassium or activation from potash feldspar [12,13]. These include high-temperature calcination, hydrothermal chemical reaction [14], hydrofluoric acid decomposition [15], and microbial decomposition [16,17]. However, these methods are seldom carried out for industrial production due to their high energy consumption, harsh conditions, or low economy. The potassium release in feldspar often occurs at high temperatures and in the presence of added sodium sulfate, sodium chloride, and sodium hydroxide, among others [18,19,20]. Furthermore, Liu et al. (2019) showed most methods not environmentally green, resulting in further industrial restrictions [19]. Lv et al. (2013a) demonstrated that highly intensive grinding could improve the physicochemical properties and bioavailability of potassium-feldspar rocks [21,22]. They also found that planetary ball milling significantly enhanced K-dissolution [23]. The microcrystalline grinding might also improve the released amounts of K, Na, Si, Al, Fe, and P, as well as raise the values of pH and EC in the extracting solution [24]. Zhang stated that mechanical pressings in the presence of Ca (OH)_2_ and CaCl_2_ result in potash feldspar with different characteristics of particle size distribution, and enhance the water-soluble and available potassium [25]. Alkaline hydrothermal alteration of K-feldspar can make K in secondary reaction products and the feldspar interface [26]. Nevertheless, the effects of mechanical force in the presence of sodium hydroxide and salts at ordinary temperatures have not yet been reported. The calcination process was used to decompose potassium feldspar and form potassium chloride using different kinds of salts, such as CaCl_2_, NaCl, and CaSO_4_ [27].

In this paper, the changes and mechanisms of soluble and available potassium induced by mechanical activation and the addition of sodium hydroxide/salts at ordinary temperatures were studied along with the roles of their ions. The effects of adding admixtures on the diffraction mechanism of microcrystalline potassium feldspar were evaluated by XRD to analyze the mechanochemical effects on the structure and efficacy of potassium feldspar powder, as well as explore the direct preparation of mineral soil conditioners by mechanical force to provide a theoretical basis for the more effective use of water-insoluble potassium minerals.

## 2. Materials and Methods

### 2.1. Raw Materials

Raw potassium feldspar powder was provided by Lingshou County Shengpeng Mineral Factory, Shijiazhuang City, China. Its main chemical composition is shown in Table 1.

The particle size distribution of raw powder consisted of 3.70% (<0.15 mm, 100 mesh), 47.86% (0.15~0.50 mm), and 48.44% (0.50~1.00 mm).

The additives consisted of analytical reagent NaOH (Beijing Beihua Fine Chemicals Co., Ltd., Beijing, China), NaCl (Modern Oriental Technology Development Co., Ltd., Beijing, China), and Na_2_CO_3_ and Na_2_SO_4_ (Tianjin Hengxing Chemical Reagent Manufacturing Co., Ltd., Tianjin, China).

### 2.2. Methods

Firstly, for detecting whether alkali of additive and processing time affect the particle size distribution, crystal structure and potassium availability, potassium feldspar powder was ground with different additives (NaOH, NaCl, Na_2_CO_3_, and Na_2_SO_4_) at a certain mass percentage using self-developed mechanical activation equipment (WJH-01,Tsinghua University, Beijing, China). The equipment WJH-01 was a kind of ball mill which can turn at high speed. The grinding bodies were steel balls whose diameters were within 5~8 mm. The grinding speed was 150 r/min. The samples were then taken out at 15, 30, 45, 60, 90, 120, 150, 180, 240, and 300 min. All samples were tested for particle size distribution and then dried at 105 ℃ for chemical analysis and other measurements. The XRD samples were sieved with a sieve of 200-mesh to remove large, gathered particles.

Secondly, the effects of additives on soluble potassium of feldspar between the grinding process and simple mixing degree were studied and the results were compared to prove the effect of the base during mechanical grinding not induced by mixing time. The raw feldspar powder, the dried sample ground for 3 h, and their respective mixtures with NaOH at 5% mass were shaken for 0, 15, 30, 45, 60, 90, 120, 150, 180, 240, and 300 min at a water: sample ratio of 10:1, respectively. The dissolved potassium content after filtration was measured three times. Among the tests, the 0 min treatment corresponded to a filtering treatment after momentary shaking. The experimental procedure is schematically displayed in Figure 1. 

### 2.3. Characterization

An ARL Perform X-ray fluorescence spectrometer (ThermoFisher, Waltham, MA, USA) was used to determine the elemental composition of the raw potassium feldspar powder. A SmartLab X-ray diffractometer with Cu-K-Ni filtered radiation (Rigaku, Tokyo, Japan) was employed for X-ray diffraction measurements of the samples in the 2θ scanning range of 10° to 90° and at a speed of 8.2°/min. After passing through a 320-mesh sieve, the sample was filled in a slot and flattened with a glass plate and placed in the sample chamber for testing, then continuously scanned by a closed copper target radiation source with the voltage 40 kV and the current 40 mA. The obtained XRD patterns were then analyzed with Jade software (MDI jade 6).

A laser particle size analyzer (9300-S, Dandong Bettersize, Dandong, China) was utilized for the direct determination of particle size distribution (PSD). 

The morphology and microstructure were identified by a field emission scanning electron microscope (SEM, KYKY-EM6200, Beijing China). All the feldspar powder samples were dried at 120 °C in a drying oven and observed with SEM. The energy dispersive spectroscope (EDS) (OXFORD INSTRUMENT, Oxford, UK) associated with SEM was used for characterization of the chemical composition of the raw potassium feldspar powder and its activated powder with sodium hydroxide.

The water-soluble potassium contents of the feldspar samples were determined three times by a filtrate-based flame photometer after shaking together with water for 30 min at a ratio of 1:10. The determination of available potassium consisted of shaking the ground samples mixed with 2 mol/L nitric acid solution at a ratio of 20:1 of liquid: sample for 30 min, filtering, and determining the potassium content of the filtrate with flame photometry [28]. The tests were repeated three times. 

The obtained data were then analyzed by a statistical package of SPSS (SPSS software, Version 17.0, SPSS Institute Inc., New York, NY, USA) [29]. Here, all treatment effects were determined by Duncan’s multiple-range testing, and significant treatment effects were presented at *p* < 0.05. 

## 3. Results

### 3.1. Change in Particle Size Distribution of Feldspar Subjected to Mechanical Grinding

Here, D_50_ represented the average particle size of a powder sample and D_97_ denoted the particle size of a coarse sample [21]. The changes in particle size distribution of potash feldspar were directly affected by mechanical grinding, as shown in Figure 2. The average particle size (D_50_) of feldspar illustrated a rapid decrease during the first 45 min, followed by a gentle decline. The addition of NaOH, Na_2_CO_3_, NaCl, and Na_2_SO_4_ slowed the decline in D_50_ of potassium feldspar due to the viscosity of the solution, which reduced the grinding effect. Moreover, some fine particles became dissolved and eventually disappeared, leading to increased average particle size.

As shown in the particle size distribution diagram (Figure 3a), many fine particles with diameters of less 1 μm were produced after 45 min grinding and became agglomerated after 180 min. However, when sodium alkali or salt was added, the particle sizes declined during the whole 300 min due to the dispersing effect of sodium ions, leading to no agglomeration (Figure 3b–f). Moreover, sodium hydroxide may have contributed to the dissolving effect of a few fine particles with a diameter of less than 1 μm (Figure 3b,c). There were few fine particles with a diameter of less than 0.5 μm occurring in the presence of Na_2_SO_4_ (Figure 3f). The difference of the particle size distribution diagram is very strange and interesting.

### 3.2. Influence of Additives on Water-Soluble Potassium Contents of Mechanically Ground Feldspar Powder

The water-soluble potassium contents of feldspar treated with additives and control treatment obtained during mechanical activation at ordinary temperatures are provided in Figure 4. The water-soluble potassium of all treatments increased as a function of activation duration. The water-soluble potassium content of the control treatment was low, but increased steadily within 300 min, from 21.64 mg kg^−1^ to 596.55 mg·kg^−1^. The water-soluble potassium contents of feldspar with added NaOH at 5% W and 10% W increased rapidly for 60 min, from 21.64 mg kg^−1^ to 1101.58 mg·kg^−1^ and 2145.95 mg·kg^−1^, respectively, followed by a slow and steady rise. Hence, the activation of mechanical force only enhanced the energy of a few potassium ions, allowing them to leave the feldspar particles. The binding of feldspar to potassium was still strong, and sodium hydroxide could act on feldspar under mechanical grinding conditions to move more potassium ions into aqueous solution. Furthermore, the soluble potassium of feldspar with added 10% NaOH reached 3044.08 mg·kg^−1^, a value higher than that of 5% W (1495.81 mg·kg^−1^). As a result, hydroxide ions played an important role in raising soluble potassium.

The integration mechanism between NaOH and potassium feldspar may relate to the substitution effect of Na^+^ for K^+^ or the chemical reaction of NaOH with fine potassium feldspar particles. To clarify this hypothesis, further experiments were carried out by adding Na_2_CO_3_, NaCl, and Na_2_SO_4_ with equal sodium of 5% as a mechanical activating assistant at ordinary temperatures. As shown in Figure 4, the water-soluble potassium of feldspar treated with several sodium salts was also higher than that of the control but lower than the treatment performed with NaOH (same Na^+^ amount of substance) for 300 min. Therefore, the mechanical grinding facilitated the substitution of some potassium ions by sodium ions, as well as enhanced the mechanochemical reaction between potassium feldspar and sodium hydroxide. This, in turn, promoted the dissolution of potassium feldspar to produce more K^+^ in the aqueous solution when compared to the case with added sodium salts. Differences in released potassium at various added sodium salts were also observed. The water-soluble potassium contents were enhanced nearly linearly in the presence of NaCl and Na_2_CO_3_. However, the increase in water-soluble potassium in the presence of Na_2_CO_3_ treatment tended to be flat after grinding for 240 min. The water-soluble potassium of feldspar treated with Na_2_SO_4_ illustrated a rapid reaction process during the initial stage but tended to flatten after 150 min. In this case, water-soluble potassium content at 300 min was also lower than those obtained with other additive treatments. Consequently, the salts anions also played a role in the potassium release of potassium feldspar under mechanical grinding conditions, especially after 150 min. In addition to the exchange of ions, the mechanochemical reaction between salts and potassium feldspar may have also occurred, especially after long-term mechanical processing.

The mathematical simulation equations at various water-soluble potassium contents and grinding durations are summarized in Table 2. The changes in soluble potassium versus the grinding time can be fitted by the linear equation: *y* = 59.64 + 1.83*x*, with *R*^2^ = 0.9431. In presence of NaOH (5% and 10% mass), the power function equation *y* = 418.98(*x* − 0.01)^0.23^ can be used with *R*^2^ = 0.9751, as well as *y* = 808.25*x*^0.23^ with *R*^2^ = 0.9613, respectively. The difference in simulation equations indicated changes in reaction kinetics of potassium dissolution in the presence of sodium alkali.

### 3.3. Dissolution of Potassium Feldspar in Water and Sodium Hydroxide Solution

To investigate whether the release of potassium from feldspar in the presence of NaOH was related to a simple chemical reaction or mechanical activation, potassium release from feldspar in water and NaOH solution under mechanical shaking conditions was tested, and the results provided in Figure 5. Most soluble potassium ions could dissolve rapidly from raw feldspar powder into water and increase slowly during shaking. The water-soluble potassium in raw potash feldspar during 300 min shaking rose from 21.64 mg·kg^−1^ to 34.23 mg·kg^−1^. Similar results were also observed for the raw feldspar powder in NaOH solution. The water-soluble potassium of the raw potash feldspar with added 5% NaOH during 300 min shaking rose from 30.94 mg·kg^−1^ to 49.09 mg·kg^−1^. The rising rate of dissolved potassium from the feldspar to NaOH solution was the same as that in water during shaking. The dissolution speed of potassium in the feldspar sample mechanically activated for 180 min can be divided into three periods: very quick (not observed), intermediate (0–60 min), and slow. The water-soluble potassium of the activated potash feldspar increased from 62.26 mg·kg^−1^ to 110.49 mg·kg^−1^ during 300 min shaking. The water-soluble potassium of the active potash feldspar with added 5% NaOH rose from 143.59 mg·kg^−1^ to 162.93 mg·kg^−1^ during 300 min shaking. Some potassium ions dissolved very quickly in water at the start then followed by dissolving at intermediate speed during the first 60 min. Afterward, the water-soluble potassium increased slowly, meaning that mechanical grinding produced some highly active potassium ions and half-highly active potassium ions. The latter were perhaps still absorbed by feldspar particles. However, the half-highly active potassium ions dissolved in the NaOH solution very quickly. The amount of potassium dissolved from mechanically active feldspar in the NaOH solution was higher than in water.

All the time, the potassium dissolved from mechanically activated feldspar far exceeded that from raw feldspar during shaking. Comparison of Figure 4 and Figure 5 suggests that the water-soluble potassium of feldspar ground treated with sodium hydroxide far exceeded the potassium of mechanically activated samples dissolving in NaOH solution. Thus, mechanochemical action between feldspar and sodium hydroxide occurred during grinding.

### 3.4. Effect of Additives on Available Potassium Content in Mechanically Ground Feldspar

A high correlation existed between the potassium extracted by 2 mol/L cold nitric acid and the potassium supplied to crops by the soil [28]. Note that potassium extracted by 2 mol/L cold nitric acid was named available potassium [14]. As shown in Figure 6, the available potassium content of each treated feldspar sample was significantly higher than that of the water-soluble potassium. During grinding, the available potassium content of feldspar increased greatly compared to the water-soluble potassium. The available potassium content of feldspar can be raised by adding sodium hydroxide at 5% of feldspar mass for 300 min to 3896 mg·kg^−1^, nearly equal to the results at 300 min of the control sample with no additives. In other words, the addition of sodium hydroxide at 5% feldspar mass only accelerated the release of potassium during the first stages, but did not enhance it at longer periods. The available potassium in the treated sample in the presence of 10% NaOH was 5162.53 mg·kg^−1^, far more than other treated samples, meaning that the available potassium of feldspar may be raised further by increasing the proportion of NaOH.

However, except for the sample treated with sodium hydroxide, whose available potassium increased rapidly compared to the control in a power function within 300 min, the available potassium of other treatments showed a linear increase during 300 min, nearly equal to or less than the control (Figure 6 and Table 3). As a result, the grinding induced mechanochemical action between feldspar and sodium hydroxide, thereby facilitating the exchange of H^+^ with K^+^ in feldspar. The available potassium in the control and some other treated samples incremented in direct proportion to the grinding duration or the consumption of mechanical energy. The available potassium content increased due to changes in the structure of potassium feldspar caused by transfer from mechanical energy. The increase in available potassium in the treated samples by sodium hydroxide was also related to mechanochemical effects. As shown in Figure 6, the available potassium contents in samples treated with Na_2_CO_3_ and Na_2_SO_4_ were slightly lower than that of the control. This may be due to the adsorption of CO_3_^2−^ and SO_4_^2−^ with strong electrical properties on the solid surface during grinding, resulting in the formation of a solid water film reducing the damaging effect of the mechanical force on the potassium feldspar structure. Note that sodium salts could only replace the potassium atoms in feldspar particles’ surfaces to promote the water solubility of potassium. However, their failure to enhance the internal structure damage of potassium feldspar resulted in no increase in available potassium.

The mathematical simulation equations for various available potassium contents and grinding durations are summarized in Table 3. The changes in available potassium with grinding time can be fitted by the linear equation *y* = 391.07 + 12.06*x*, with *R*^2^ = 0.9963. In the presence of NaOH (5% and 10% mass), the power function equations *y* = 435.48(*x* + 1.15)^0.38^ with *R*^2^ = 0.9837, as well as *y* = 1112.69(*x* − 0.03)^0.27^ with *R*^2^ = 0.9810 can be used, respectively.

### 3.5. Changes in Specific Surface Area of Feldspar Powder during Grinding

Atoms in the solid surfaces are more active than inner ones due to their less bonding [30]. Thus, the samples treated with NaOH or NaCl at 5% of feldspar mass were selected for the measurement of specific surface areas. As shown in Figure 7, the mechanical grinding could increase the specific surface area of potassium feldspar. Moreover, the specific surface areas of samples with additives were lower than that of the control sample after grinding for 45 min. The main reason for this had to do with the viscosity of the solution, which increased in the presence of additives. The pulverization effect of the particles was also reduced. Another reason might have to do with soluble salt additives that may cement the particles and block the microcracks during the drying process, thereby declining the measured values of the specific surface areas.

The addition of sodium hydroxide dissolved some fine particles, as well as blocked microcracks by soluble matters during the drying treatment. Consequently, the specific surface area of the potassium feldspar powder with NaOH reached a maximum value of 3.799 m^2^·g^−1^ at 180 min. Afterward, the specific surface area decreased and remained below the specific surface area of the control without additives. 

### 3.6. Influence of Mechanical Grinding on Crystal Structure of Minerals in Potassium Feldspar

The XRD diffraction intensity of the material showed a unit cell in the crystal grain, positively correlated with the crystallinity [31]. The changes in diffraction intensities of the main minerals of feldspar during mechanical grinding are gathered in Figure 8 and Table 4. The XRD diffraction intensities of all three minerals showed a continuous decreasing trend during grinding. The XRD diffraction intensities declined in the following order: SiO_2_ > NaAlSi_3_O_8_ > KAlSi_3_O_8_. The decline in diffraction intensities of the minerals meant the creation of crystal defects and a decrease in crystallinity, indicating variations in the inner structure of the feldspar powder. Such changes facilitated the reaction of more potassium ions with other ions, thereby raising the available potassium. This transformed potassium feldspar crystals into microcrystals, or even amorphous fine particles.

The variations in diffraction intensities of mechanically ground potassium feldspar treated with 5% NaOH are shown in Figure 9 and Table 5. The XRD diffraction intensities of all three minerals displayed a downward trend as a function of grinding time. Moreover, the changing trends looked the same as those of the control samples. However, the XRD diffraction intensities of NaAlSi_3_O_8_ (002) crystal plane and SiO_2_ (011) crystal plane in samples treated with NaOH decreased significantly during grinding when compared to those without NaOH treatment. The XRD diffraction intensities of NaAlSi_3_O_8_ (002), KAlSi_3_O_8_ (220) crystal plane, and SiO_2_ (011) crystal plane in samples treated with NaOH decreased most during first 0~15 min, 15~90 min, and 90~180 min, respectively, compared to that without NaOH treatment. This indicates the feldspar mineral can be affected more easily than the silica mineral by NaOH, but the latter was affected more deeply than the former since it reacted with sodium hydroxide during the mechanical grinding. The XRD diffraction intensity of KAlSi_3_O_8_ (220) crystal plane decreased rapidly at the early stage followed by rising from 90 min to 180 min and then falling again. This may be induced by stress release.

Above all, the mechanical force played the leading role in declining the mineral crystallinity, and silica also helped the decline.

A similar changing trend as that of the samples treated with NaOH was also observed for the XRD diffraction intensities of potassium feldspar powder treated with NaCl and Na_2_CO_3_ (Table 6). In particular, the crystal plane (220) diffraction intensities of the KAlSi_3_O_8_ mineral showed an increasing phase in all three samples. This can be explained by the effect of stress relief when the crystal structures of SiO_2_ and NaAlSi_3_O_8_ minerals were broken, and the potassium feldspar gained a new stable state after changing. 

### 3.7. Impact of Mechanical Grinding on Particle Surface of Potassium Feldspar

The chemical action always started from a solid surface, and potassium ions were released from the feldspar through its surface. In Figure 10, the mechanical activation did not only reduce the particle size of potassium feldspar but also induced rough and uneven particle surfaces covered by many nanoparticles (Figure 10e, curve corresponding to 90 min). This greatly increased the specific surface area. After mechanical processing for 150 min, some nanoparticles aggregated, and fractal composite particles occurred. Since the chemical reaction of solid particles often started from the surface, more potassium atoms on the surface can be exchanged by other ions to be released into the soil solution, thereby raising the available potassium in potash feldspar. Furthermore, the nanoparticles possessed small particle size and high activity; thereby they not only react easily with soil solution or soil particles but also may be directly absorbed by various organisms like soil microorganisms, plant roots, and protozoa, becoming direct sources of nutrients like potassium and silicon for organisms.

The variations in particle appearance of potassium feldspar ground with sodium hydroxide are illustrated in Figure 11. Overall, particles became smaller after 150 min with some dissolved by sodium hydroxide solution, thereby changing the appearance of particles. Under high magnification (×50,000), nanoparticles were observed adhering to bigger particles ground for 15 min. However, the nanoparticles of the sample ground for 90 min became fewer due to dissolution in alkali liquor, such as SiO_2_ + OH^−^ = H_2_SiO_3_^−^, KAlSi_3_O_8_ + 2H_2_O + 6OH^−^ = 3H_2_SiO_4_^2−^ + Al (OH)_4_^−^ + K^+^, and NaAlSi_3_O_8_ + 2H_2_O +6OH^−^ = 3H_2_SiO_4_^2−^ + Al (OH) _4_^−^ + Na^+^ [2] (Production & International Trade and Agriculture Services., 2017–2021, Wang, 2006). As consumption of alkali and further grinding increased, the particles became rougher.

The changes in the grain size of potassium feldspar treated with NaCl during mechanical grinding are shown in Figure 12. The particles were gradually broken to become smaller, resulting in big blocks breaking into smaller ones with all shapes and sizes. Moreover, the particles gathered to form fractal structures.

The potassium feldspar treated with Na_2_CO_3_ changed more obviously during the mechanical activation process (Figure 13). Moreover, although bigger particles became smaller, as under other treatments, the sizes of smaller ones looked homogeneous when compared to samples treated with NaCl.

The particle sizes of potassium feldspar treated with Na_2_SO_4_ became smaller, leading to agglomerations to form bigger ones (Figure 14). 

The main elements on the feldspar particle surface were described by EDS (Figure 15). The potassium ions in the presence of NaOH were released more from the surfaces of the feldspar particles when compared to the control after processing (Spectrum1). The sodium ions in the presence of NaOH entered the surface of the feldspar particles through ion exchange, resulting in higher sodium content than the control (Spectrum2).

## 4. Discussion

### 4.1. Change in Particle Distribution and Crystal Structure of Potassium Feldspar by Mechanochemical Activation

Potassium feldspar is a solid composed of complex components, which was chipped to become smaller after mechanochemical grinding. The action of tear would produce new sections, leading to potassium feldspar crystal lattices with broken bonds, dislocations, and uneven textures. Cation exchange along the newly formed crack flanks produced Na-enriched diffusion halos around the cracks, and the associated lattice contraction and tensile stress state caused continuous crack growth [32]. These features changed the XRD peaks. The milling activation step induced structural changes in flying ash to promote its reactivity in alkaline solution and produced morphology [33]. The mechanochemical activation resulted in potassium feldspar with finer particles of different sizes, where surfaces of bigger particles became adhered to finer ones to result in fractal structures [34]. Agglomerated particles within two size ranges were formed in mechanochemically treated muscovite, a small fraction between 0.1 and 1.0 μm [35]. The rougher K-feldspar surfaces exhibit increased Cm (III) uptake and stronger complication [36].

The addition of sodium hydroxide or its salts affected the production and aggregation of fine particles, where the former could dissolve fine particles. Some fine particles possibly consisted of salt crystals under SEM [37], indicating that the reaction between potassium feldspar and sodium hydroxide can be controlled by solid product layer diffusion. The decrease in particle size might abate the diffusion resistance to increase the reaction rate and cut down the activation energy [1]. The mechanical grinding can not only reduce the reactant particle diameters but also accelerate the reaction process, leading to a mechano-chemical effect. Prior mechanical activation of the K-feldspar led to substantial increases of up to 172% in K release [22,38].

### 4.2. Acceleration of Ion Exchange by Mechanochemical Activation

Most potash feldspars belong to orthoclase with a three-dimensional framework body composed of [AlO_4_] tetrahedrons or [SiO_4_] tetrahedrons sharing a vertex with the connected The pores of the framework are filled with ions of alkali metal and alkaline-earth metal located in gaps among TO_4_ tetrahedrons, such as K^+^, Na^+^, Ca^2+^, and Ba^2+^ [1]. The alteration was faster in amorphous oligoclase than in its crystalline equivalent due to the more open structure of the glass [39]. Hydrothermal alteration of K-feldspar, i.e., the hydrolytic dissolution of K-feldspar framework coupled with the incorporation of Ca^2+^ in place of K^+^ [14]. The mechanical grinding did not only break the crystal structure but also reduced the particle size and enhanced the specific surface area. The K^+^ amount on the surface of the particles increased to become easily soluble due to no cancellation inside. Moreover, some inner K^+^ was soluble as shown by the solution curve ascending as a function of vibration time. The addition of Na^+^ could substitute the K^+^ absorbed by anions. The potassium extraction kinetics in the presence of both additives was satisfactorily corroborated by the Ginstling–Brounshtein model [1]. Ion exchange between solution cations and lattice K^+^ is observed only for fully flexible slabs, where the release of K^+^ is facilitated by lattice vibrations. The exchange rates are strongly surface-dependent [40]. On the other hand, sodium hydroxide as a strong base dissolved some [SiO_4_] particles and broke down some of the three-dimensional framework, thereby increasing soluble potassium under mechanical grinding or shaking. However, Wang et al. (2006) found that the increase in OH^−^ concentration in solution strengthened the dissolution reactions of albite and potash feldspar at 298 K, with the former case greater than the latter [37]. Hydrogen ions with a smaller diameter and stronger substitution ability led to more extracted potassium from potassium feldspar by 2 mol/L HNO_3_ when compared with those obtained by sodium salts. The increase in specific surface area and produced inner fissures by mechanical force-induced hydrogen ions easily seep inside potassium feldspar, followed by replacement of potassium ions. Hence, the soluble and available potassium increased due to surface area expansion and structural change. These data were not all consistent with the declining trend of potassium feldspar particle size.

## 5. Conclusions

Sodium salts and hydroxide were employed during mechanical grinding to improve the release of potassium from feldspar such as the water-soluble and available potassium of samples taken at different times. The causes of the phenomenon were analyzed by particle distribution, crystal construction, and surface area measurements. The action of mechanical force was evaluated by the contrast of mechanically shaking the potassium feldspar in water and NaOH solution. Several important conclusions are drawn.

(1) Mechanical grinding made the potassium feldspar particles with smooth surfaces become fine and rough. However, the use of additives curbed the production of submicron particles with diameters of 0.1~1 μm. Meanwhile, the presence of sodium hydroxide dissolved some fine potassium feldspar particles.

(2) Mechanical grinding and additives increased the water-soluble potassium of feldspar. The water-soluble potassium increase in feldspar was not only caused by the particle finess. The mechanical grinding also broke some feldspar crystals and transformed them into microcrystal or amorphous forms, which could be characterized with the changes in XRD peak intensities. On the other hand, the water-soluble potassium of feldspar was enhanced by additives due to the ion exchange effects of mechanical grinding. Sodium hydroxide could dissolve some fine potassium feldspar particles, thereby raising the content of water-soluble potassium, and so it was more than that obtained from sodium salts. The advancement in mechanical activation looked different from the simple physical shaking effect, with the former looking stronger than the latter. This difference indicated that grinding based on mechanochemistry and mechanical energy facilitated the release of potassium from feldspar.

(3) The available potassium of feldspar can be increased by mechanical force. This can also be accelerated by sodium hydroxide within 300 min, but not raised by sodium salts. The energy from the mechanical activation coupled with the addition of sodium hydroxide led to changes in the structure of potassium feldspar, making the inner potassium ions easy to extract by HNO_3_. The addition of sodium salts produced little effect on the structure of feldspar, thereby barely influencing its available potassium content. 

## Figures and Tables

**Figure 1 materials-17-00144-f001:**
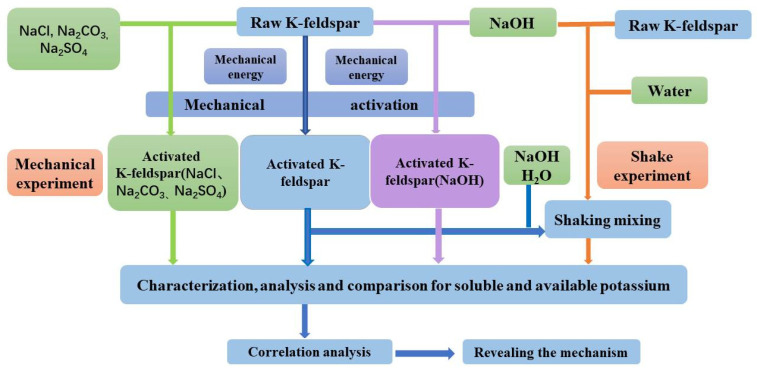
Experimental design sketch.

**Figure 2 materials-17-00144-f002:**
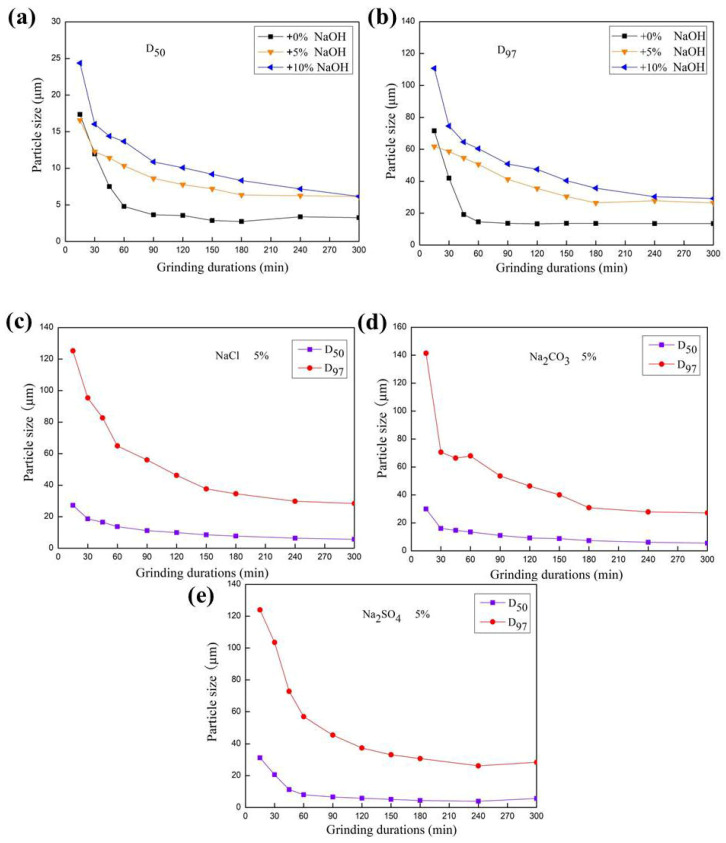
Changes in particle size (D_50_, D_97_) of potassium feldspar in presence of additives during grinding. (**a**) D_50_ in presence of no additive, 5% NaOH and 10% NaOH, (**b**) D_97_ in presence of no additive, 5% NaOH and 10% NaOH, (**c**) D_50_ and D_97_ in presence of 5% NaCl, (**d**) D_50_ and D_97_ in presence of 5% Na_2_CO_3_, (**e**) D_50_ and D_97_ in presence of 5%Na_2_SO_4_.

**Figure 3 materials-17-00144-f003:**
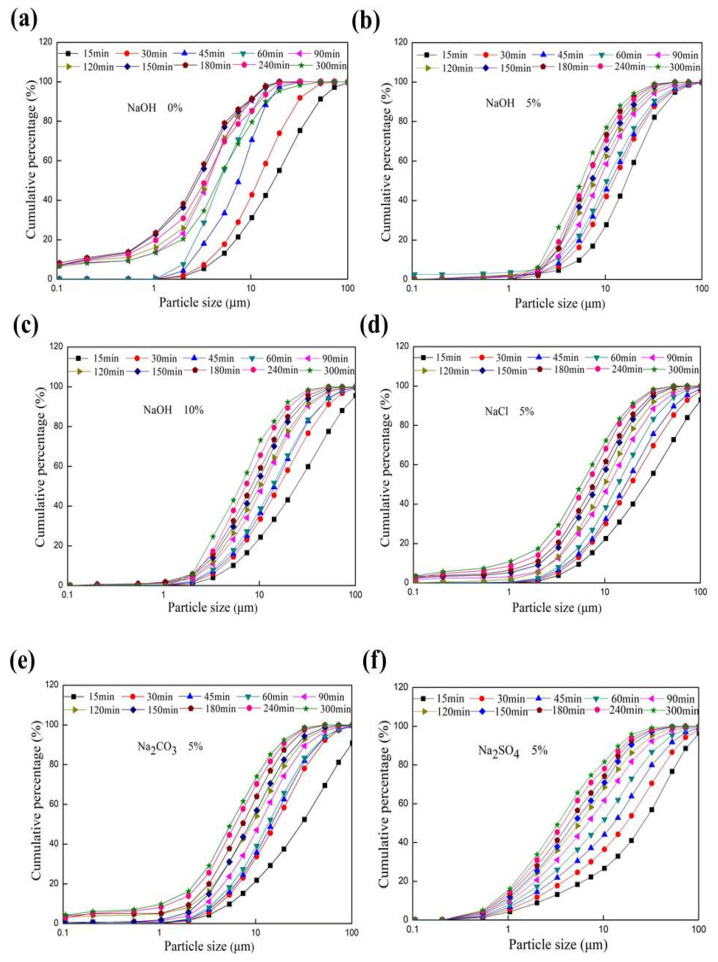
Change in the PSD of potassium feldspar in the presence of additives during grinding. Additives: (**a**) no additive, (**b**) 5% NaOH, (**c**) 10% NaOH, (**d**) 5%NaCl, (**e**) 5% Na_2_CO_3_, and (**f**) 5% Na_2_SO_4_.

**Figure 4 materials-17-00144-f004:**
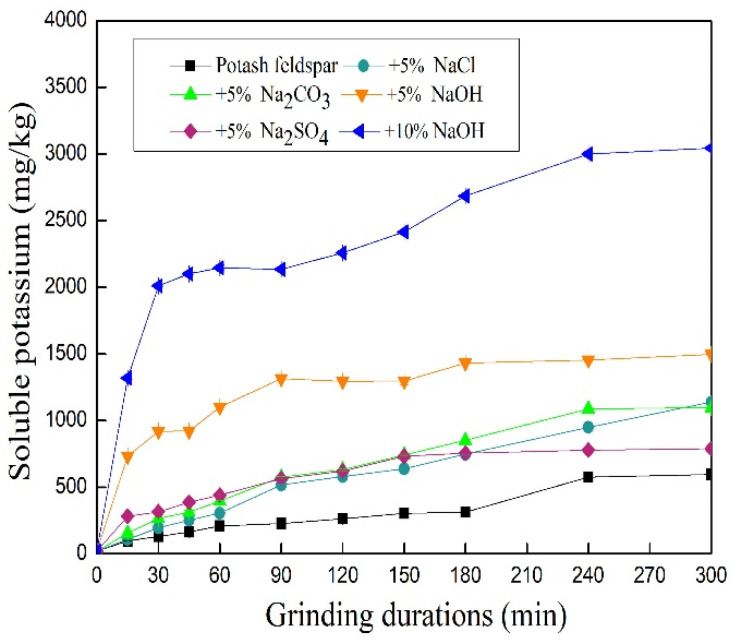
Changes in water-soluble potassium contents of feldspar in presence of NaOH and sodium salts under mechanical activation (reported values are statistically significant (*p* < 0.05)).

**Figure 5 materials-17-00144-f005:**
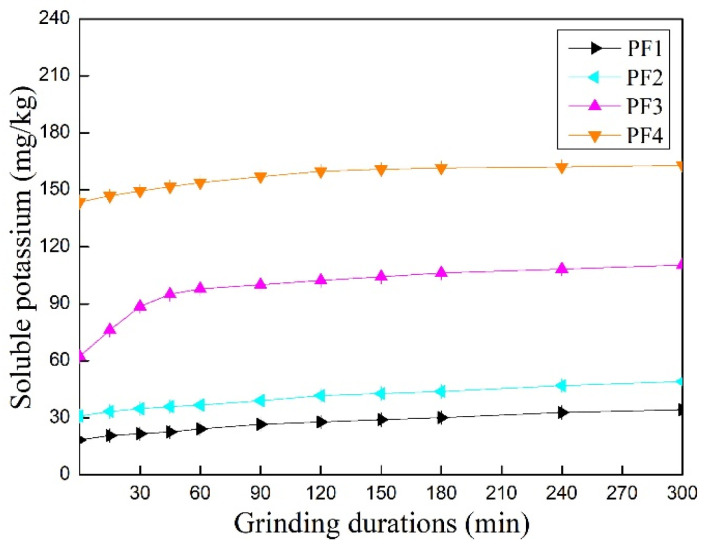
Potassium dissolved in pure water and NaOH solution during shaking: raw powder (PF1), raw powder + NaOH (5%) (PF2), ground feldspar for 180 min (PF3), and ground feldspar for 180 min + NaOH (5%) (PF4) (Reported values are statistically significant (*p* < 0.05)).

**Figure 6 materials-17-00144-f006:**
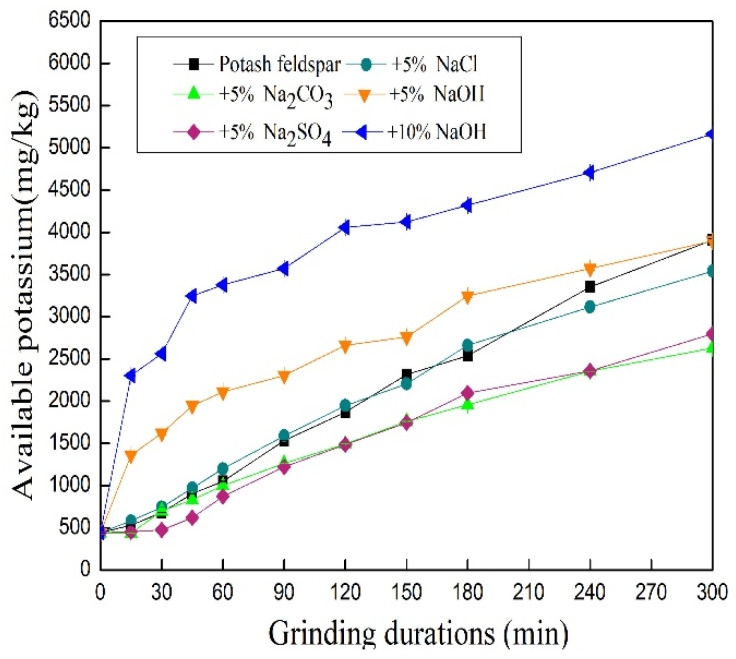
The changes in available potassium of feldspar in the presence of additives during grinding (reported values are statistically significant (*p* < 0.05)).

**Figure 7 materials-17-00144-f007:**
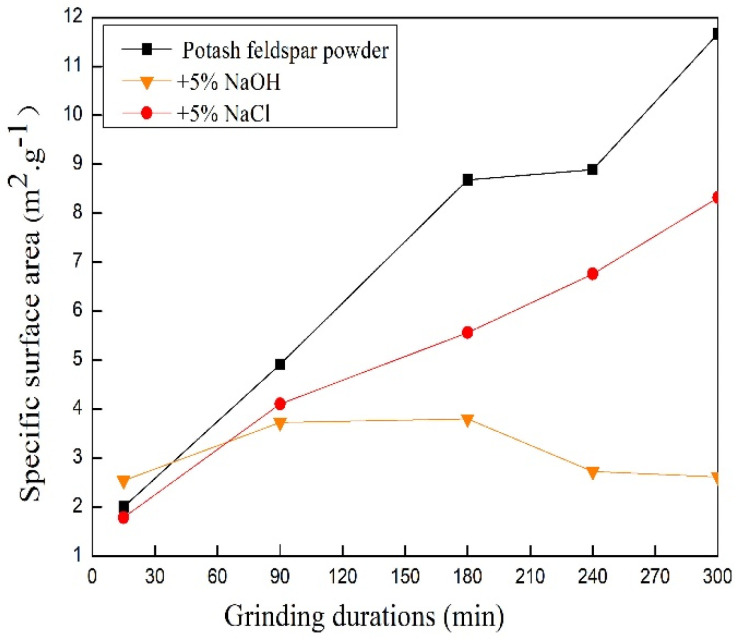
The changes in specific surface area as a function of grinding duration.

**Figure 8 materials-17-00144-f008:**
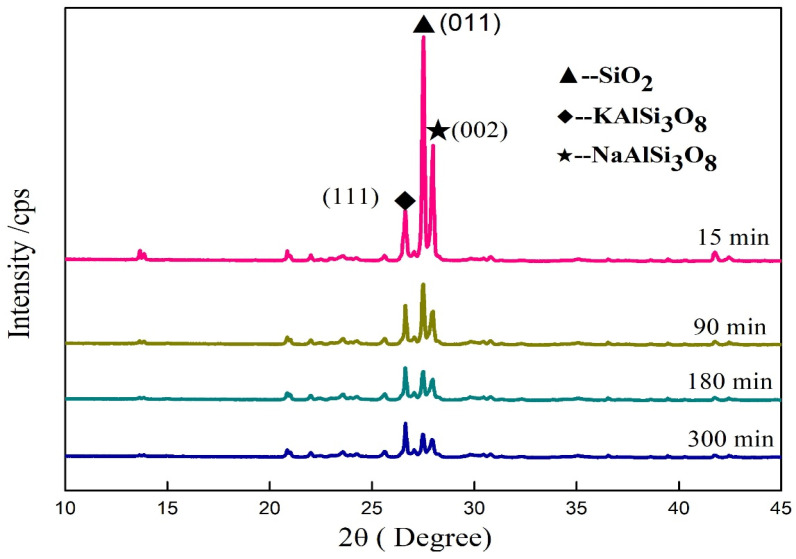
Changes in XRD diffraction peak intensities of potassium feldspar during grinding.

**Figure 9 materials-17-00144-f009:**
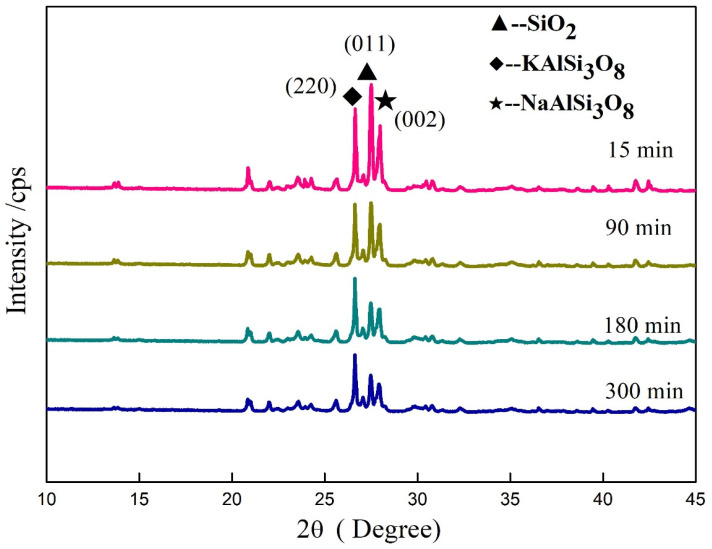
Changes in XRD diffraction peak intensities of potassium feldspar mechanically activated with 5% NaOH.

**Figure 10 materials-17-00144-f010:**
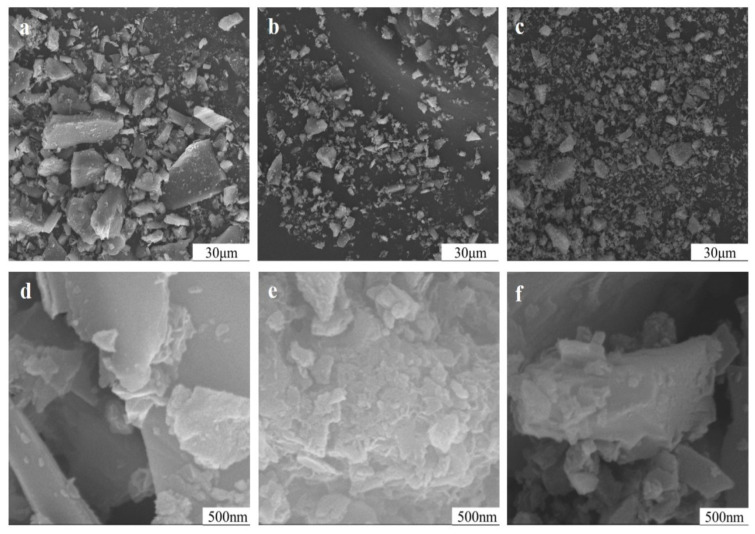
Particle morphologies of potassium feldspar powder at different grinding durations: (×1000) 15 min (**a**), 90 min (**b**), and 150 min (**c**), (×50,000) 15 min (**d**), 90 min (**e**), and 150 min (**f**).

**Figure 11 materials-17-00144-f011:**
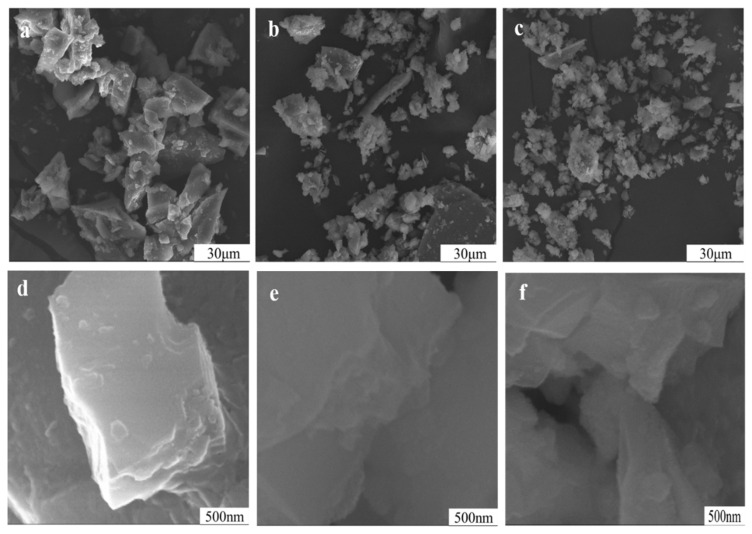
Particle morphology of potassium feldspar powder treated with 5% NaOH at different grinding times of: (×1000) 15 min (**a**), 90 min (**b**), 150 min (**c**), (×50,000) 15 min (**d**), 90 min (**e**), and 150 min (**f**).

**Figure 12 materials-17-00144-f012:**
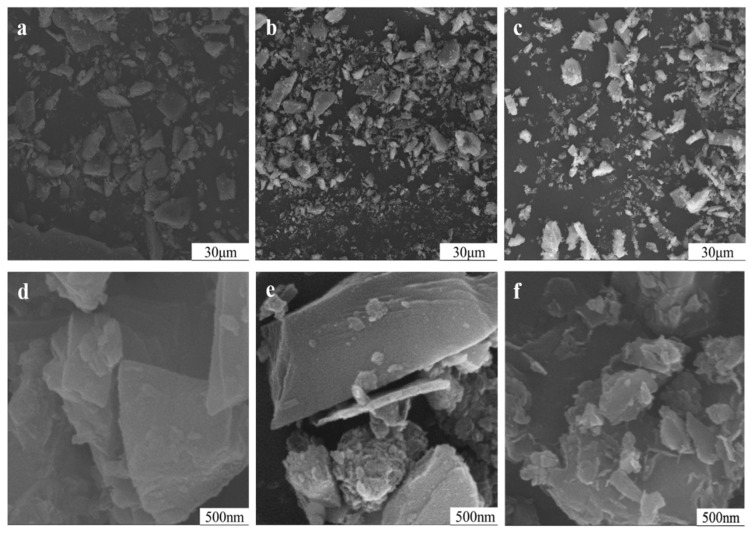
The particle morphologies of potassium feldspar powder treated with NaCl at different grinding durations: (×1000) 15 min (**a**), 90 min (**b**), 150 min (**c**), (×50,000) 15 min (**d**), 90 min (**e**), and 150 min (**f**).

**Figure 13 materials-17-00144-f013:**
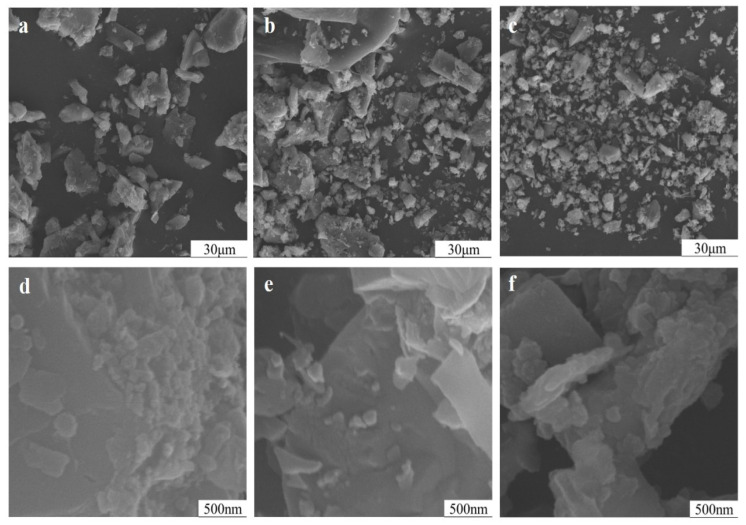
Particle morphologies of potassium feldspar powder treated with Na_2_CO_3_ for different grinding times: (×1000) 15 min (**a**), 90 min (**b**) 150 min (**c**), (×50,000) 15 min (**d**), 90 min (**e**), and 150 min (**f**).

**Figure 14 materials-17-00144-f014:**
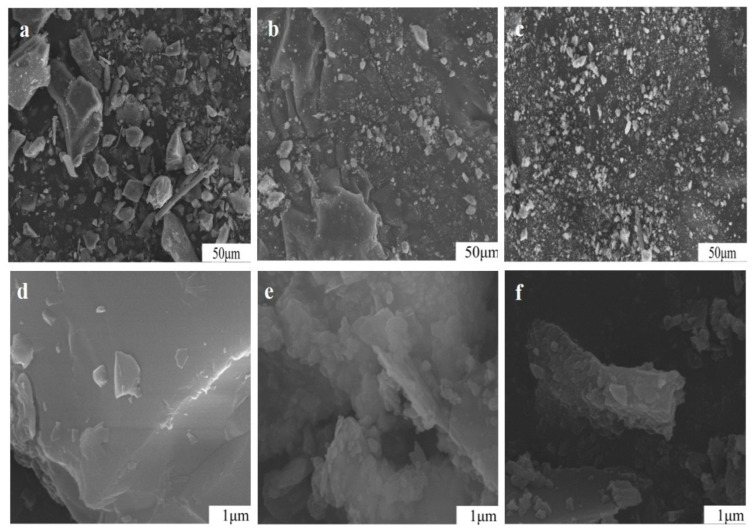
Particle morphologies of potassium feldspar activated by mechanical force with Na_2_SO_4_ for different grinding times of: (×500) 15 min (**a**), 90 min (**b**), 150 min (**c**), (×25,000) 15 min (**d**), 90 min (**e**), and 150 min (**f**).

**Figure 15 materials-17-00144-f015:**
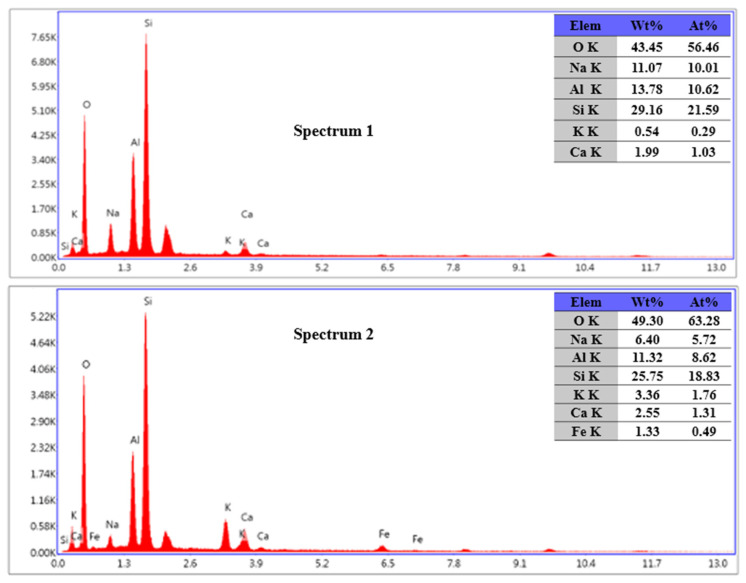
Energy dispersive X-ray spectroscopy (EDS) images of potassium feldspar powder and their elemental compositions.

**Table 1 materials-17-00144-t001:** Total elements of potash feldspar powder.

Element	Content (%)	Element	Content (%)	Element	Content (%)	Element	Content (%)
SiO_2_	67.03	CaO	1.44	Rb_2_O	0.0293	S	0.0139
Al_2_O_3_	18	BaO	0.633	Cr_2_O_3_	0.0273	Nd_2_O_3_	0.0040
K_2_O	8.69	MgO	0.106	Ag_2_O	0.0113	PbO	0.0039
Na_2_O	3.36	P_2_O_5_	0.0708	MnO	0.0054	Ga_2_O_3_	0.003
Fe_2_O_3_	0.359	TiO_2_	0.0323	NiO	0.0024		
SrO	0.155	Cl	0.018	ZnO	0.0011		

**Table 2 materials-17-00144-t002:** Fitting equations of water-soluble potassium of feldspar activated by mechanical force.

NaOH Addition	Fitting Equation	*R* ^2^	Fitting Equation Type
0% mass	*y* = 59.64 + 1.83*x*	0.9431	linear equation
5% mass	*y* = 418.98(*x* − 0.01)^0.23^	0.9751	power function equation
10% mass	*y* = 808.25*x*^0.23^	0.9613	power function equation

**Table 3 materials-17-00144-t003:** Fitting equation of available potassium of feldspar activated by NaOH at various grinding durations.

NaOH Addition	Fitting Equation	*R* ^2^	Fitting Equation Type
0% mass	*y* = 391.07 + 12.06*x*	0.9963	linear equation
5% mass	*y* = 435.48(*x* + 1.15)^0.38^	0.9837	power function equation
10% mass	*y* = 1112.69(*x* − 0.03)^0.27^	0.9810	power function equation

**Table 4 materials-17-00144-t004:** Changes in XRD diffraction intensities of potassium feldspar during grinding.

Time/min	The Diffraction Peak		
Mineral	KAlSi_3_O_8_	SiO_2_	NaAlSi_3_O_8_
2θ	26.64°	27.54°	28.00°
Crystal plane	220	011	002
0	14,546	51,322	23,928
15	11,950	15,713	10,404
90	9624	9858	6803
180	9211	8382	5794
300	7944	6273	4708

**Table 5 materials-17-00144-t005:** Changes in XRD diffraction intensities of feldspar treated with 5% NaOH during grinding.

Time/min	Diffraction Peak Intensity		
mineral	KAlSi_3_O_8_	SiO_2_	NaAlSi_3_O_8_
2θ	26.64°	27.54°	28.00°
Crystal plane	220	011	002
0	14,546	51,322	23,928
15	11,864	15,479	9447
90	8975	9213	6105
180	9351	5712	4892
300	8233	5259	3998

**Table 6 materials-17-00144-t006:** XRD diffraction intensities of potassium feldspar treated with 5% NaOH, NaCl, and Na_2_CO_3_ during grinding.

Time/min		The Diffraction Peak Intensity		
Mineral	KAlSi_3_O_8_	KAlSi_3_O_8_	KAlSi_3_O_8_	KAlSi_3_O_8_
Additives		+NaOH	+NaCl	+Na_2_CO_3_
2θ	26.64°	26.64°	26.64°	26.64°
crystal plane	220	220	220	220
0	14,546	14,546	14,546	14,546
15	11,950	11,864	11,083	9207
90	9624	8975	9643	10,776
180	9211	9351	9943	8959
300	7944	8233	8329	7647

## Data Availability

Data are contained within the article.

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
