# Peer review of "Mechanism of Potassium Release from Feldspar by Mechanical Activation in Presence of Additives at Ordinary Temperatures"

_materials, 2023, doi:10.3390/ma17010144_

Round 1
Reviewer 1 Report
Comments and Suggestions for Authors
Authors performed research on changes and mechanism of soluble and available potassium induced by mechanical activation and addition of sodium hydroxide/salts at ordinary temperatures
Good introduction, supported by adequate references.
Aims clearly stated.
Adequate Materials presentation.
Methodological approach is adequate and correct, but needing more information (including Norms/Protocols/References); for instance XRD is not presented on Materials subchapter
Results presentation is, in general, clear and sound.
Discussion is well supported by data and figures/tables, but must be improved comparing obtained results with more similar ones (from other authors)
Conclusions are coherent with obtained results.
Comments on the Quality of English LanguageMinor editing
Reviewer 2 Report
Comments and Suggestions for Authors
The manuscript entitled ‘Mechanism of Potassium Release from Feldspar by Mechanical Activation in Presence of Additives at Ordinary Temperature’ raises the very important issue of using mechanochemical activation of minerals to increase potassium availability from natural sources. However, the article also demonstrates some drawbacks that need to be corrected before the publication of this manuscript.
Section 2.2 Methods : ‘using self-developed mechanical activation equipment (WJH-01).’ - please explain what the equipment is, and give the parameters of the grinding process, i.e. grinding speed, grinding material and bowl.
Figure 15 - please improve the resolution, for which exact samples was the EDS spectrum measured? indicate from which picture and from which point the chemical composition was measured.
Please complete the latest literature on mechanochemistry and mechanochemical activation of different materials and minerals, e.g. https://doi.org/10.1016/j.cemconres.2022.106962 ; https://doi.org/10.3390/ma15207174 ; https://doi.org/10.1016/j.conbuildmat.2022.128739
Reviewer 3 Report
Comments and Suggestions for Authors
This is an interesting paper with a topic of interest for this journal. To improve the quality of the article, the following aspects should be taken into account:
- Figure 1 is not clear as to its reading sequence. It is suggested to present guidelines as lines that facilitate the reader's interpretation.
- It is suggested that the authors present each of the stages of the experimental design in a clearer way. Make it clear what each of the processes carried out to achieve the objective consists of, since it is not completely clear what was done.
- The mechanisms that occur during the different phases of the work should be explained more clearly. In some sections, the explanations of the processes should be more detailed and not only descriptive. The role of the different additives used should be explained. Why do they work? What is the mechanism of action?
- Page 12 presents two figures and their explanation is very limited.
- Increase the number of references corresponding to the last 5 years.
